# Airstream Association of Large Boundary Layer Rolls during Extratropical Transition of Post-Tropical Cyclone Sandy (2012)

James A. Schiavone 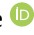

Independent Researcher, Bridgewater, NJ 08807, USA; jimschiavone@gmail.com

**Abstract:** Better understanding of roll vortices that often occur in the tropical cyclone (TC) boundary layer is required to improve forecasts of TC intensification and the granularity of damaging surface winds. It is especially important to characterize rolls over a wide variety of TCs, their environments, and TC development phases. Boundary layer rolls have been observed in TCs since 1998, but only recently in a TC during its extratropical transition phase. The work reported herein is the first to analyze how boundary layer rolls are distributed among the extratropical features of a transitioning TC. To this end, routine and special operational observations recorded during landfalling Post-tropical Cyclone Sandy (2012) were leveraged, including radar, surface, rawinsonde, and aircraft reconnaissance observations. Large rolls occurred in cold airstreams, both in the cold conveyor belt within the northwestern storm quadrant and in the secluding airstream within the northeastern quadrant, but roll presence was much diminished within the intervening warm sector. The large size of the rolls and their confinement to cold airstreams is attributed to an optimum inflow layer depth, which is deep enough below a strong stable layer to accommodate deep and strong positive radial wind shear to promote roll growth, yet not so deep as to limit radial wind shear magnitude, as occurred in the warm sector.

**Keywords:** hurricanes; boundary layer; roll vortices; landfall; extratropical transition

## 1. Introduction

Roll vortices (rolls hereafter) in the hurricane boundary layer (HBL) are organized features that consist of vertically overturning circulations which are elongated approximately in the hurricane's tangential wind direction. Rolls have been observed in the HBL since 1998 [1,2], and when present, they modulate the strength of the surface wind, typically by ±30% in wind speed [3]. Total wind is enhanced in their downdraft regions and reduced in their updraft regions, providing a mechanism for the patchy wind damage often observed under hurricanes [4]. This vertical motion also transports momentum and enthalpy between the surface and the free atmosphere of a hurricane, which plays a key role in modulating hurricane intensity. Thus, observational characterization of HBL roll vortices is important for improving forecasts of hurricane intensity and the damaging winds that hurricanes cause during landfall.

Rolls occur in atmospheric boundary layers beyond hurricanes. In fact, rolls were originally associated with airmass cloud bands [5–8]. Furthermore, rolls with different characteristics and driving mechanisms have since been identified and classified [9,10], with multiple types of rolls also identified and classified for hurricanes [2,11–17]. Large eddy simulations have been used to characterize rolls at finer granularity in recent years [18–23], while theoretical underpinning for roll formation has been developed by Mourad and Brown [24] and, specifically for hurricane environments, by Foster [25].

Hurricane Sandy made landfall on New Jersey in 2012 as a post-tropical cyclone and caused devastating storm surge flooding, widespread and long-duration power outages due to massive treefall, and 72 direct deaths in the continental United States. Although rolls recently have been observed to be associated with high wind bands in extratropical

cyclones [26], to our knowledge, rolls never have been reported as observed in a tropical cyclone during its extratropical transition phase until those recently reported for landfalling Post-tropical Cyclone (Post-TC) Sandy [27]. Because Post-TC Sandy was in its final stage of extratropical transition just before landfall, the fairly dense set of observations during landfall over the landfall region afforded the opportunity to study how its HBL rolls were associated with key features of the extratropical transitioning cyclone's synoptic and mesoscale structure.

In our prior work [27], exceptionally large rolls (5–14 km wavelengths) were reported as observed during Post-TC Sandy's landfall, and their large size was attributed to the large depth of the positive radial wind shear layer. In this work, we examine how the synoptic and mesoscale environment may have contributed to the deep positive radial shear layer and the location prevalence of the observed rolls. This is the first observational study of the relationship of roll vortices to extratropical features of a tropical cyclone during its extratropical transition. Section 2 describes the observations and the supporting simulation profile data used in this analysis. Section 3 describes the verification of the simulation profiles and develops an objective metric to quantify observed roll presence for use in studying the geographical distribution of large rolls. Section 4 proposes a cause of the observed geographical distribution of the large rolls, and Section 5 summarizes key findings.

## 2. Data and Methods

### 2.1. Observations

This work leverages the existing enhanced-density observation data collected during Post-TC Sandy's landfall phase in New Jersey. Profile data consist of hourly wind profile observations at the Fort Dix, NJ WSR-88D Doppler radar site [28], 6-h rawinsonde launches at Islip, NY [29] and several near-shore dropsondes from aircraft reconnaissance flights [30]. Surface data comprise 5-min observations from the Rutgers New Jersey Weather Network [31] and hourly oceanic observations from the NOAA National Data Buoy Center [32]. We also analyzed radial velocity scans at the Fort Dix WSR-88D radar [28] and 1-s aircraft reconnaissance observations [30] for near-shore flight legs.

Doppler radar data are from the WSR-88D radar station located at Fort Dix, NJ (blue dot labeled "DIX" in Figure 1). The center of Post-TC Sandy traversed southern New Jersey (red line in Figure 1) in a west-northwestward direction, about 50 km south of the radar site. Full-volume radar scans are produced approximately every 6 min, with a minimum elevation angle of 0.5 degree. Both reflectivity and radial velocity data were used, for which range resolutions are 1 and 0.25 km, respectively, with an azimuthal resolution of 1 degree for both. Doppler radar measures the motion of precipitation particles in the direction of the radar beam. Hereafter, we refer to the velocity observed by Doppler radar as radar radial velocity, which should be distinguished from the radial velocity of the post-tropical cyclone relative to its storm center.

To characterize the vertical wind profile during landfall, Doppler radar velocity azimuth display (VAD) wind profile data for Fort Dix were analyzed. The wind profiles are calculated by the WSR-88D VAD algorithm [33], where the mean radar radial velocity is calculated as the first harmonic of a Fourier analysis applied to the total radar radial velocity. The method is applied to each height among a predetermined set of heights above ground. Each of these heights intersects at least one radar elevation-angle scan-cone at a particular radar range. The mean radar radial velocity is then computed for each range and elevation angle pair that matches the appropriate height above ground. The result is a profile of wind speed and direction data for each radar volume scan, typically recorded every 6 min but stored digitally only hourly.

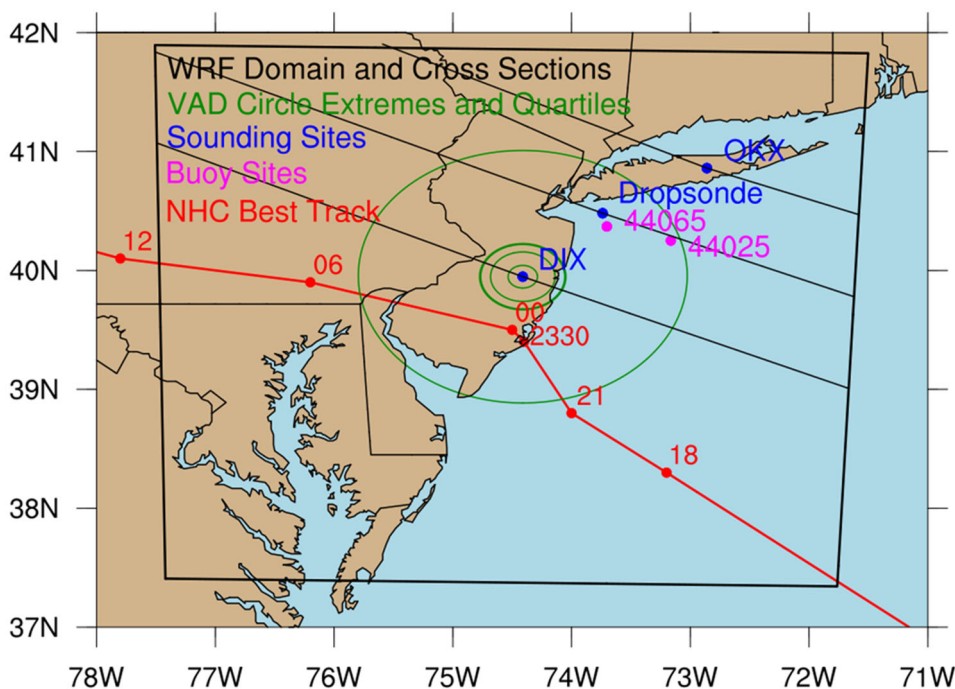

**Figure 1.** Location information for data used. The large black rectangle illustrates the extent of the 500 km × 500 km range of the Weather Research and Forecasting (WRF) simulation analyzed, and the 3 black diagonal lines locate vertical cross-sections analyzed for the simulation. Green ovals denote the quantile and extrema circles of the radar velocity azimuth display (VAD) wind profile observations. Three blue dots are sounding sites, where DIX is the Fort Dix WSR-88D radar, OKX is the Islip rawinsonde site, and Dropsonde is the analyzed launch at 2136 UTC 29 October 2012. Magenta dots are buoy sites, and red dots and lines represent the National Hurricane Center (NHC) Best Track of Sandy.

The radar profile speed and direction data are subsequently converted to radial and tangential wind components relative to Sandy's storm center and its translation vector. (Since the storm was classified as a hurricane or a post-tropical cyclone during different periods of certain analyses, a storm-type prefix for Sandy will be eliminated to avoid unwieldy verbiage for these cases.) Storm center coordinates were specified using best track data from Blake et al. [34]. Because of the relatively coarse temporal resolution of the observed track (intervals are at most 6 h), storm center coordinates were calculated for every 6-min radar scan time by linear interpolation between best track storm center positions. The storm center translation vector was calculated for each best track storm center position as a mean vector among 3 track positions centered on a specified location. Analogous to the method used for storm center coordinates, translation vectors were also interpolated for each radar scan time. Finally, storm-centered cylindrical coordinates were used to compute the radial and tangential wind components relative to the translating storm center. Since the profiles are calculated for the circular intersections of scan-cones and height-planes, they represent mean velocities for circles of various radii centered on the radar site at Fort Dix. The middle 50% of the distribution of these radii is in the relatively narrow radii range of 23–31 km, although the extrema reach 10 and 117 km. The radii for the quartiles and extrema are depicted as green ovals in Figure 1.

Other observations include 3 rawinsonde launches at Islip on central Long Island [29] which were analyzed for 1200 and 1800 UTC 29 October 2012 and 0000 UTC 30 October 2012. (Unless otherwise noted hereafter, all times are on 29 October 2012 except 0000 UTC, which is on 30 October 2012.) Moreover, 8 near-shore legs of aircraft reconnaissance flight observations [30] north of the storm center were analyzed between 1900 and 0000 UTC. These 1-s interval observations were recorded at an elevation of about 1 km above the ocean. The lone dropsonde profile that was analyzed was the only one that was within the

geographical and temporal scope of this analysis and was deployed near the mouth of the Raritan River at 2136 UTC. Finally, surface observations recorded at 5-min intervals on the Rutgers New Jersey Weather Network [31] were used, which is hereafter referred to as the NJ Mesonet. Surface observations at hourly intervals for 2 oceanic stations of the NOAA National Data Buoy Center [32] network were also used: Stations 44065 and 44025, which were 25 and 72 km east-southeast of Sandy Hook, NJ, USA, respectively.

Because of the broad disparity in observation types and their differing scopes and granularity, it is useful to compare their characteristics in one concise place. To that end, Table 1 compares the spatial and temporal ranges and resolutions of the observational data used. The radar scans and NJ Mesonet data are the most comprehensive in that they both provide complete coverage of the state of New Jersey at relatively high temporal and geographical resolution. Buoy observations are limited to 2 sites but are used to provide data at key oceanic locations. Radar VAD, rawinsonde, and dropsonde observations are limited to single sites and relatively coarse time resolution, but they provide vertical profile observations.

**Table 1.** Comparison of geographical and temporal characteristics of observational data.

| Instrument | Horizontal Range | Vertical Range | Geographical Resolution | Temporal Range | Temporal Resolution |
|---|---|---|---|---|---|
| Radar scans | All of NJ | 5° cone | 1 km × 0.25° | 1200–0600 UTC | ~6 min |
| NJ Mesonet | All of NJ | Near surface | ~20 km | 1200–0600 UTC | 5 min |
| Buoys | 2 sites | Near surface | 2 sites | 1200–0600 UTC | 1 h |
| Radar VAD | 1 site | 0–5 km | 1 site | 1200–0600 UTC | 1 h |
| Rawinsonde | 1 site | 0–5 km | 1 site | 1200–0000 UTC | 6 h |
| Dropsonde | 1 site | 0–1 km | 1 site | 1 event | 1 event |

### 2.2. Archived Weather Research and Forecasting (WRF) Simulation

Archived output from a 500 m resolution WRF simulation was used to fill the gaps in the spatial continuity of the observations, particularly for profile observations. Because archived output was used, it was not possible to do any numerical experiments with the WRF simulation nor modify any of the boundary or surface layer parametrizations. Although the fixed resolution of 500 m is capable of manifesting rolls, it is not ideally capable of representing their finer scale characteristics since the model resolution resides in the "grey zone" of turbulence simulation [35–37], wherein the largest turbulent eddies are explicitly represented, but smaller ones are parameterized. Nevertheless, signatures of rolls are exhibited by the WRF simulation, whose characteristics agree reasonably well with those of rolls observed by radar [27]. Therefore, simulation results are used only to examine the mesoscale and synoptic scale environments encompassing the regions where rolls occur.

The 96-h simulation was conducted in 2013 using the Advanced Research WRF (version 3.3.1) initialized at 1200 UTC 26 October 2012 [38] using a horizontal resolution of 500 m. A single domain of size 2660 km × 2500 km (5320 × 5000 grid points) with 150 vertical layers (25 layers below 3 km height) was used without any nests. The time step was 1 s, and initial and boundary conditions were generated from the National Centers for Environmental Prediction Global Forecast System (GFS) model initialized at 1200 UTC 26 October 2012, with boundary conditions processed and applied every 6 forecast hours.

The Yonsei University (YSU) planetary boundary layer (PBL) scheme [39], the Noah land model [40], and the MM5 Monin–Obukhov similarity theory [41] surface layer model were used. Although the WRF physics was from a 2013 version, key physics for rolls is the PBL parameterization, which in our case was YSU. We are confident that YSU is viable for our analysis since it continues to be used in WRF, at grid resolutions similar to that used herein, to study small-scale vortices in HBLs (e.g., Wu et al. [42]) as well as for realistic representations of tropical cyclone (TC) surface wind fields [43–45]. Cloud physics was modeled using WSM6 6-class microphysics with graupel [46]. Convection parameterization

was not used for this high-resolution simulation. Three-hourly outputs of the data set used herein are archived [47]. The 500 km square area analyzed herein is denoted by the large black rectangle in Figure 1.

## 3. Results

### 3.1. WRF Profiles Verification

Prior work [27] validated the WRF simulation's track and timing of Post-TC Sandy. This work validates WRF profiles using observed Fort Dix, NJ WSR-88D wind profiles and Islip, NY rawinsonde profiles. No comparison is made with the dropsonde profile because its depth is limited to the lowest 1 km above the ocean, and only a single dropsonde observation is at a location and time useful for this analysis.

Beyond a few exceptions at specific times and heights, the WRF simulation profiles agree reasonably well with profile observations at Fort Dix and Islip during the main analysis period of 1800 through 0000 UTC. Figure 2 shows profiles of storm-center-based radial and tangential wind components (blue and orange, respectively) and radial wind shear (gray) at Fort Dix and Islip, as located on the map in Figure 1, calculated for both observed and simulated winds (solid and dashed, respectively). Tangential winds agree fairly well in magnitude and profile shape, although altitudes of peak winds differ because of the complex shapes of the profiles. Radial winds agree well at 0000 UTC (Figure 2c,e), but at earlier times, the WRF simulation's maximum inflow below 2 km is 10–20 m/s weaker than observed (Figure 2a,b,d). Radial wind shear profiles agree reasonably well, considering the substantial noise in the observed profiles.

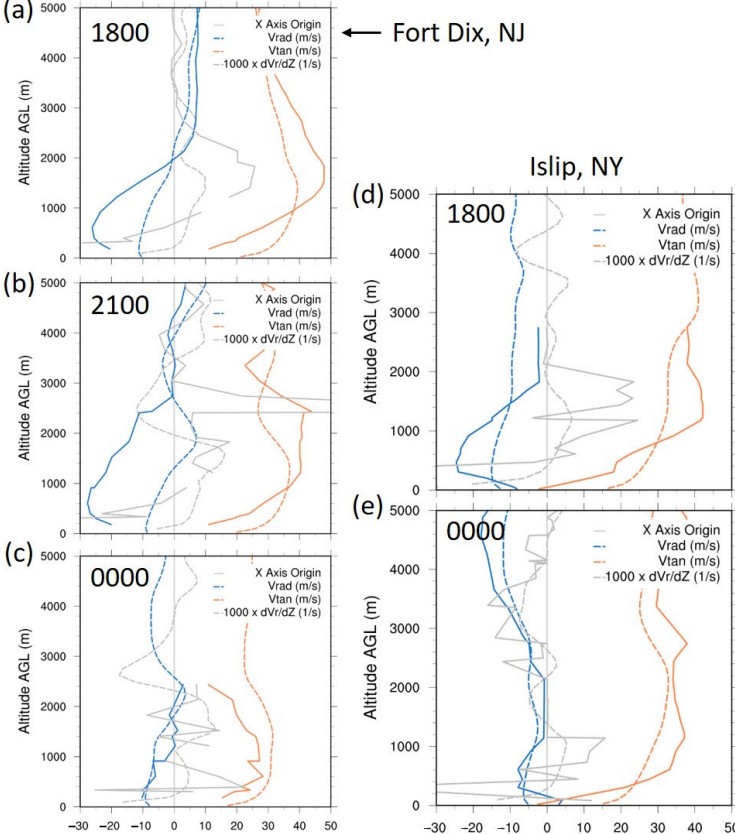

**Figure 2.** Comparison of observed and simulated wind profiles. Observed and WRF-simulated profiles are shown as solid and dashed lines, respectively. Blue and orange are radial and tangential wind components, respectively, in m/s units, and grey is 1000 times the vertical gradient of radial wind in s$^{-1}$ units. (**a–c**) Radar VAD and WRF-simulated winds at Fort Dix at 1800, 2100, and 0000 UTC, respectively, (**d**,**e**) rawinsonde and WRF-simulated winds at Islip at 1800 and 0000 UTC, respectively. All wind components are relative to the storm center.

Temperature and mixing ratio profiles at Islip are also compared (not shown) for soundings observed at 6-h intervals, 1200, 1800, and 0000 UTC, and there is good agreement except for small differences at a few specific times and elevation intervals. Temperature profile exceptions are at 1200 UTC when WRF is a few degrees warmer throughout the profile and at 1800 UTC below 1 km where WRF is warmer. Mixing ratio profile exceptions are at 1800 UTC above 2 km where WRF is dryer and at 0000 UTC above 4 km where WRF is moister. The reasonable agreement, albeit with isolated small exceptions, lends confidence in using WRF cross-sections spanning Sandy's northern airstreams to illuminate possible environmental impacts on roll occurrence, intensity, and location.

*3.2. Roll Presence Metric*

In prior work, we demonstrated that rolls were manifested in radar reflectivity and velocity fields as linear features whose spatial variation exhibited wavelengths ranging from 5 to 14 km in Sandy (Figure 3a–c shows examples from radar velocity fields). In this work, a method is developed to map observed roll presence and intensity using fields of the standard deviation of radar velocity (SDRV). Raw radar radial velocity data for the lowest radar scan angle of 0.5 degree is used without any adjustment for storm location and translation. The metric is validated by a manual visual comparison of maps of the metric with Fort Dix WSR-88D radar reflectivity and velocity imagery showing locations of rolls, similar to those shown in Figure 3a–c. Standard deviation must be calculated over an area larger than roll scale (~15 km) but smaller than storm scale (~30 km) variations. Analyses were done over an area spanning New Jersey and environs for 225 nearly-constant-area boxes of size 0.2-degree latitude and longitude, which corresponds to box dimensions of about 17 km east-west × 22 km north-south. Some results are shown in Figure 3d–f for times preceding, during and after warm sector residence over northern New Jersey, as discussed in Section 3.3 below. Roll presence metric results are blocked out near the radar site since radar velocity variations are inherently large near the radar site.

The observed warm sector presence is exhibited by the red areas in Figure 3g–i, which are shaded contour maps of the NJ Mesonet surface temperatures at about the same times as the roll presence maps of Figure 3d–f. This observed qualitative diminution of rolls in the warm sector is examined more fully and quantitively below.

*3.3. Association of Rolls with Extratropical Airstream Sectors*

Observations alone, including radar velocity and reflectivity and NJ Mesonet wind and temperature measurements, are used to elicit airstreams, their boundaries, and roll presence and intensity. Standard deviation of radar radial velocity, as described above, is used as the metric of roll presence and intensity. Aircraft reconnaissance data are also used and, despite their limited coverage, demonstrate consistency with other observations. Figure 3e shows that the roll presence metric reaches a minimum value of about 1.5 m/s throughout northern New Jersey at 2200 UTC when the warm sector is over northern New Jersey. Before and after this time, the metric is above 2 m/s (Figure 3d,f), significantly so at 1806 UTC when the cold conveyor belt (CCB), embodied by the northerly flow on the storm's western periphery, is over northern New Jersey.

This is examined quantitatively using NJ Mesonet surface temperature and wind direction time series to more precisely locate storm airstream sectors. Mean observed surface temperature and wind direction for the time series (Figure 4a,b) are calculated for 5 to 8 NJ Mesonet stations in 3 separate subregions of northern New Jersey that span eastern, central, and western parts of a rectangle, within which SDRV is calculated. Subregion surface observation stations and the SDRV region rectangle are shown in Figure 4c, along with a geographical distribution map of SDRV at 2200 UTC. The subregions, labeled East, Central, and West in Figure 4a,b, are separated in the west-to-east direction because the main storm-scale flow over northern New Jersey during this time period is generally from east to west. The accelerated temperature rise (Figure 4a) and veering of wind (Figure 4b) observed at 2040 UTC in the East subregion and at 2200 UTC in the West subregion are attributed

to the surface warm front crossing those subregions. The abrupt drop in temperature at 2230 UTC in the East subregion and at 2335 UTC in the West subregion are attributed to the secluding airstream surface boundary crossing those subregions (Figure 4a). Delays in the crossing time from east to west are consistent with the observed westward warm sector motion (Figure 3g–i).

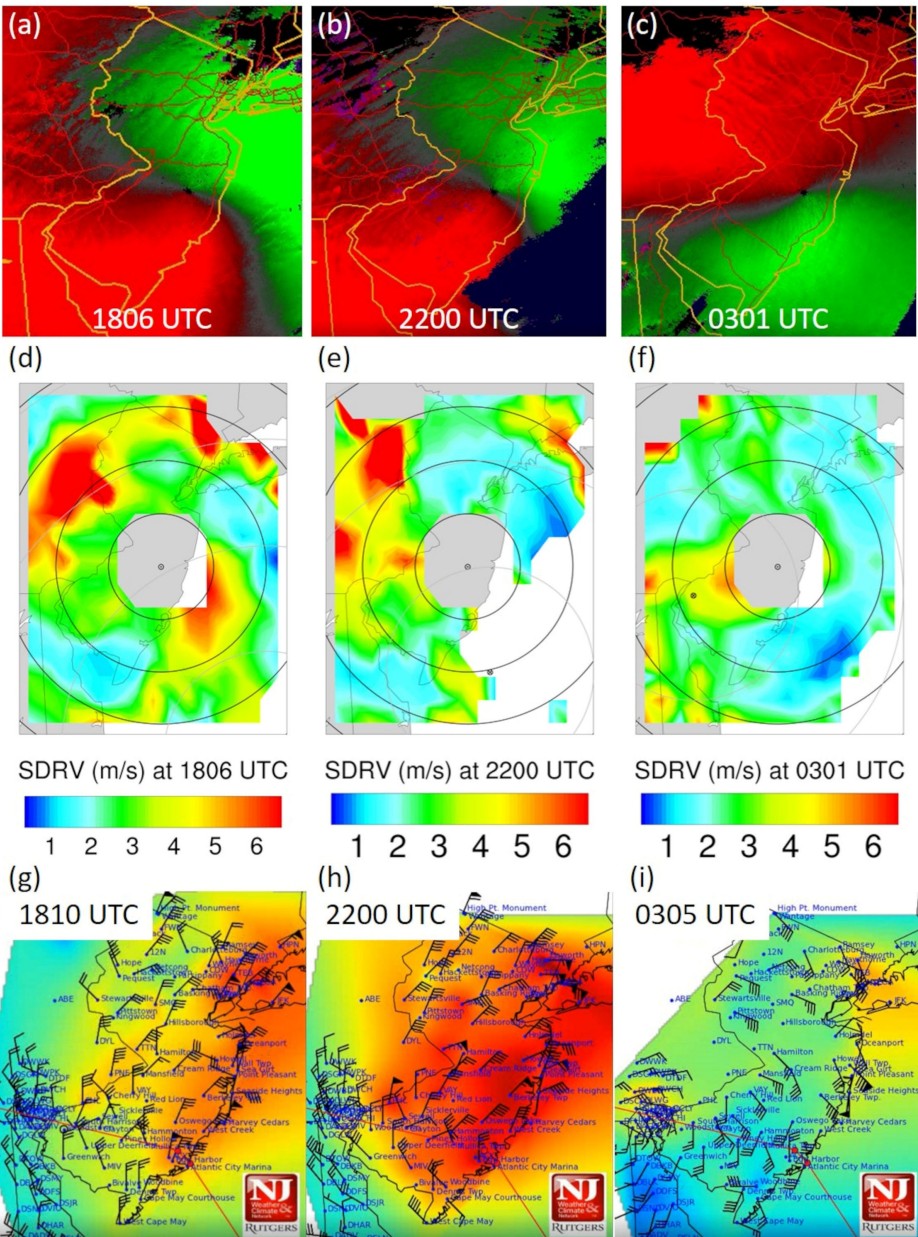

**Figure 3.** Observed radar velocity, roll presence metric, and comparison with storm airstream sectors. (**a**–**c**) Radar radial base velocity fields for a 0.5-degree elevation angle at 1806, 2200, and 0301 UTC, respectively. (**d**–**f**) Color-filled contour maps of the observed roll presence metric, standard deviation of radar velocity (SDRV), at 1806, 2200, and 0301 UTC, respectively, where 0301 UTC is on 30 October 2012. SDRV is calculated within 0.2-degree latitude/longitude squares (~17 km east-west × ~22 km north-south), with results plotted at the center of each box. The encircled dot and × locate the radar site and storm center, respectively. (**g**–**i**) Color-filled contour maps of observed surface temperature measured at NJ Mesonet stations at 1810, 2200 and 0305 UTC, respectively, where 0305 UTC is on 30 October 2012. Blue and red extrema correspond to 8 and 18 degrees C, respectively. These maps also show near-surface wind vectors as barbs, where each full flag represents 10 knots (5.1 m/s).

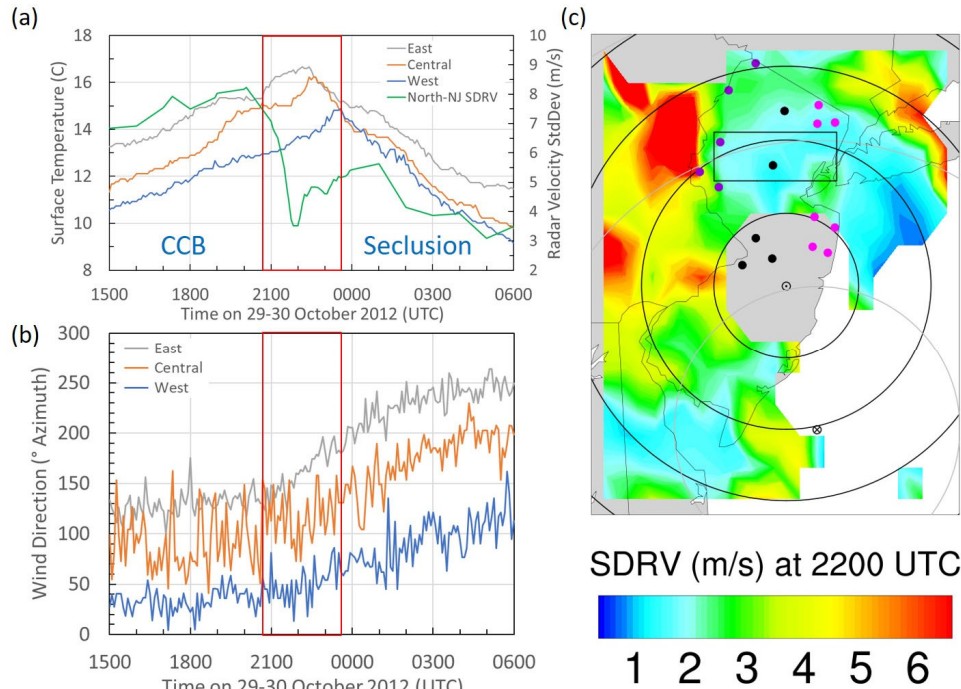

**Figure 4.** Region-averaged time series of the observed roll presence metric and surface temperature and wind direction. (**a**) Time series of surface temperature observed at NJ Mesonet stations averaged over 3 subregions of northern New Jersey (East, Central, and West), with (**c**) showing the observation sites for each subregion in magenta, black and violet dots, respectively. (**a**) Time series of the observed roll presence metric (SDRV in green) analyzed over the region enclosed by the black rectangle on (**c**). (**b**) Time series of wind direction observations averaged over the same 3 subregions of northern New Jersey as (**a**). On (**a**,**b**), time scales begin on 29 October 2012, and end on 30 October 2012 and vertical red lines bracket the warm sector's presence over northern New Jersey, with its arrival determined by the accelerated temperature rise in the East subregion and its exit by the maximum temperature in the West subregion. (**c**) SDRV observed at 2200 UTC during warm sector presence over northern New Jersey, with the encircled dot and ⊗ locating the radar site and storm center, respectively.

A time series plot of the roll presence metric, SDRV, is shown in green in Figure 4a, with its calculation domain shown as the black rectangle in Figure 4c. Figure 4a shows that, for the 3 subregions, SDRV reaches very low levels after the accelerated increases in surface temperature and before their maxima, which represents the warm sector's residence period over those 3 subregions. Thus, accelerations in temperature rise for the 3 subregions (Figure 4a) correspond in time to deterioration in roll intensity, and those deteriorations are delayed from east to west for the SDRV computed for the 3 separate subregions (not shown).

The abrupt drop in surface temperature at each NJ Mesonet station in northern New Jersey affords the opportunity to geolocate the surface seclusion boundary and its westward progression across northern New Jersey. To that end, 5-min interval surface temperature time series are used at NJ Mesonet stations in northern New Jersey to plot hourly isochrones of the surface seclusion boundary there (Figure 5a). (The surface warm frontal boundary is more diffuse and difficult to discern accurately, so it is not plotted.) Samples of the time series are shown in Figure 5b for 3 sites located roughly along the magenta line in Figure 5a, with a site near each end and one near the middle. The east-to-west delay in the arrival of the surface seclusion boundary is manifested by the corresponding delay in the surface temperature maxima. Alternatively, the seclusion boundary's progression is exhibited on the plot of its arrival time at 6 surface observation sites versus their longitude, as shown in Figure 5c. Furthermore, by comparing Figures 4c and 5a, it is evident that the warm sector, which is delineated on its eastern side by the 2200 UTC isochrone in Figure 5a, is

characterized by very low values of SDRV in that same area northwest of Raritan Bay, Staten Island and the Hudson River mouth at 2200 UTC (Figure 4c).

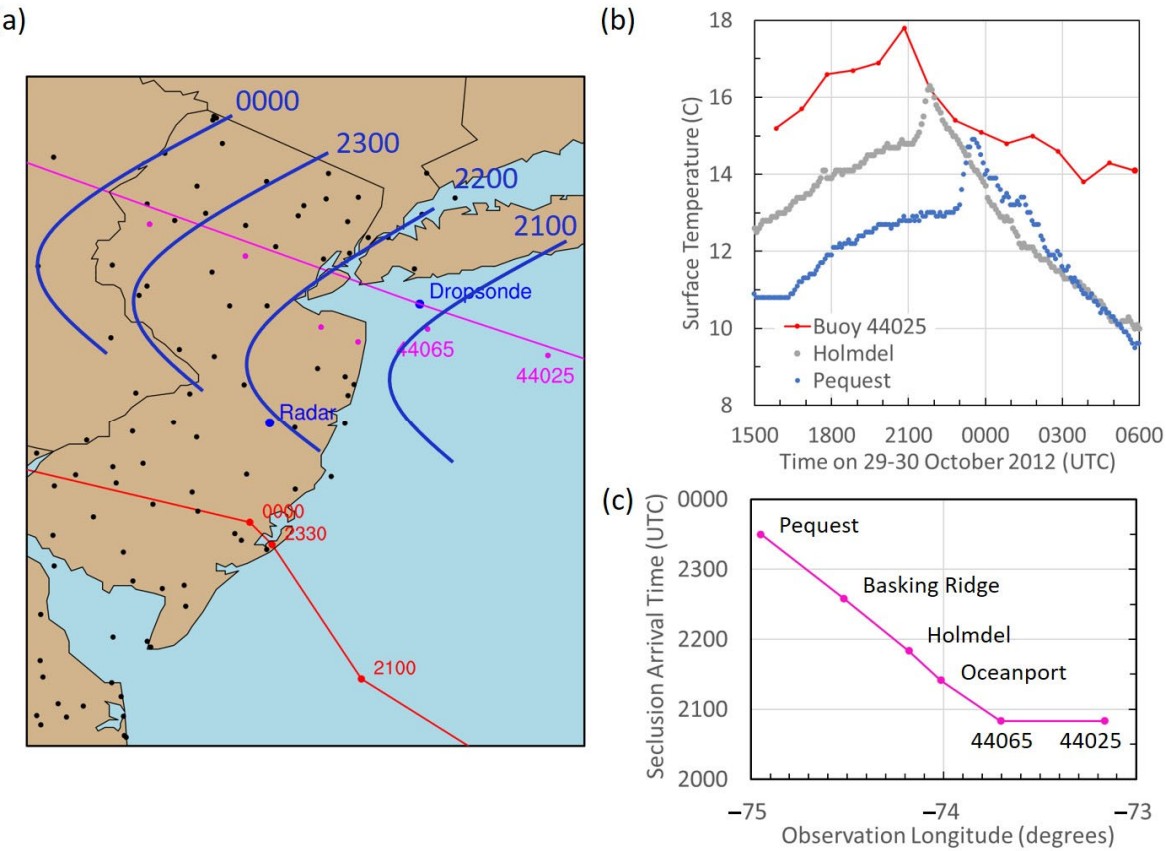

**Figure 5.** Seclusion boundary propagation west-northwestward across northern New Jersey. In (**a**) the blue-arc isochrones denote the location of the seclusion boundary at the surface at hourly intervals as derived from 5-min interval surface temperature observations at NJ Mesonet stations, where observation sites are denoted by dots on the map. (**b**) Surface temperature observations at 3 sites located at the ends and the approximate center of the magenta diagonal line on (**a**). (**c**) The seclusion boundary arrival time versus observation site longitude at 6 sites, indicated by magenta dots and lying approximately along the magenta line in (**a**). Buoys 44065 and 44025 exhibit identical arrival times because the buoy observation interval is large (hourly).

Figure 6 illustrates, as an example, the surface seclusion boundary location relative to other extratropical features of Post-TC Sandy just before landfall. The background image is of radar reflectivity at 305 m constant altitude. Rolls are manifested by the linear radar reflectivity features, which are most evident in the CCB and also present, albeit less vividly, in the seclusion airstream.

In summary, both geographical (Figure 3) and temporal (Figure 4) observational comparisons of surface temperature and roll intensity indicate that rolls are mainly confined to cold airstreams, both in advance of the warm front and within the secluding cold airstream but are essentially absent in the warm sector. Although this analysis is restricted to New Jersey because the continuity and density of wind and temperature observations are restricted to the NJ Mesonet, it is fortuitous that the final phase of the extratropical transition and its seclusion process occurred while key features of the transitioning TC traversed the NJ Mesonet.

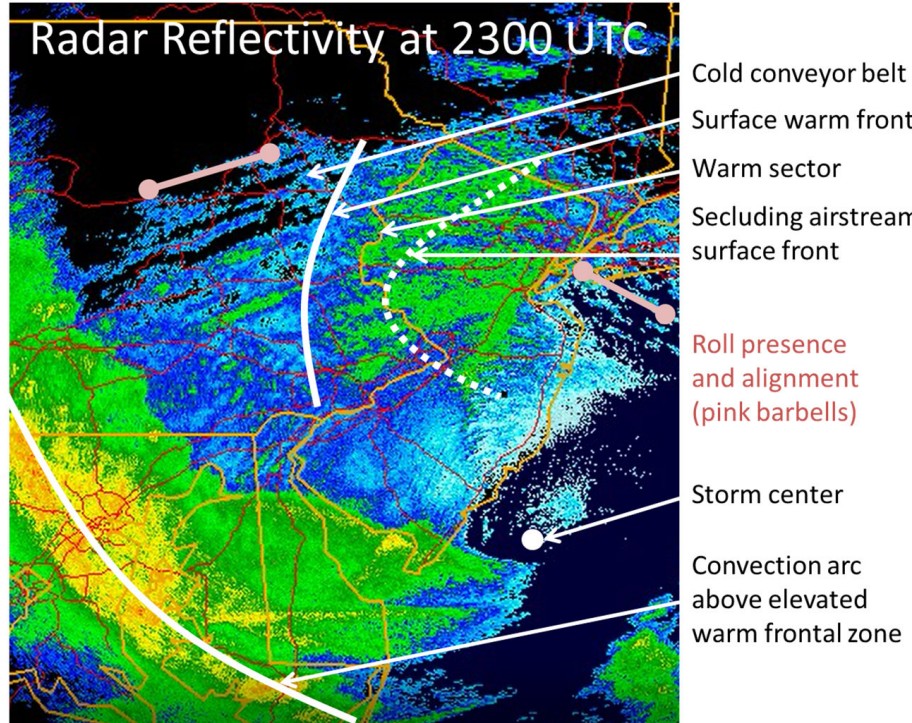

**Figure 6.** Delineation of observed structural features of Post-tropical Cyclone Sandy. Background image is radar reflectivity at 305 m constant altitude from the Fort Dix radar at 2300 UTC 29 October 2012. White symbols are as noted in the figure. Pink barbells illustrate roll regions and roll alignment.

### 3.4. Boundary Layer Characterization and Evolution

To understand both why the HBL and rolls therein were so deep and why rolls are associated almost exclusively with the cold airstreams during Post-TC Sandy's landfall, we first characterize the vertical structure and evolution of the boundary layer. Next, the parameters that drive roll growth and size are quantified and their evolution is analyzed. Finally, a hypothesis is presented for the exclusive association of rolls with cold airstreams. More rigorous testing of the hypothesis is reserved for future work, however.

We first analyze observed wind, temperature, and moisture profiles at Islip (location shown in Figure 1 as OKX), as well as their temporal evolution over the 12-h period preceding and during landfall. We also analyze vertical cross-sections normal to the frontal boundaries across northern New Jersey of WRF-simulated equivalent potential temperature ($\Theta_e$) and its vertical gradient.

Figure 7a,b shows 6-h changes in observed and simulated, respectively, profiles of potential temperature ($\Theta$) and water vapor mixing ratio ($q$) at Islip, where solid lines are for 1200–1800 UTC and broken lines are for 1800–0000 UTC. The observed profiles at Islip (Figure 7a) exhibit substantial warming and moistening during 1200–1800 UTC followed by substantial cooling and drying during 1800–0000 UTC throughout most of the atmosphere below 4 km. The WRF-simulated profiles at the Islip observation location (Figure 7b) exhibit similar behavior, although the moistening and warming during 1200–1800 UTC are shallower in the simulation. Thus, both observed and simulated profiles provide vertical structural evidence that the warm sector advanced over Islip before 1800 UTC and that the seclusion did likewise after 1800 UTC.

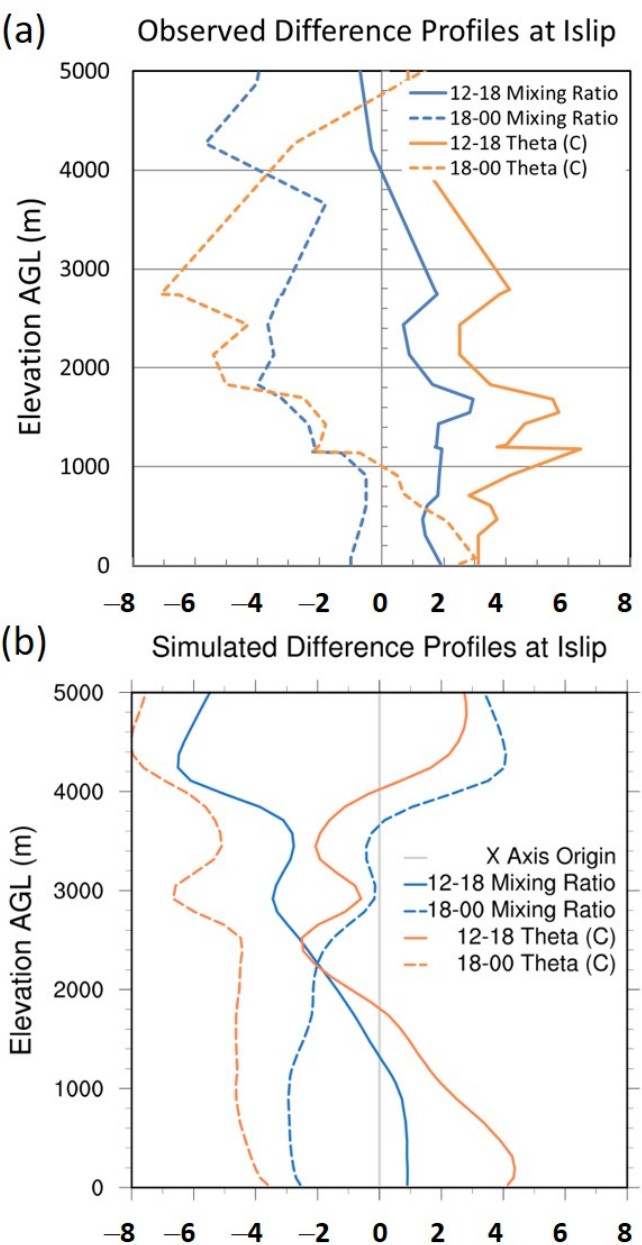

**Figure 7.** Comparison of observed and WRF-simulated profile evolution at Islip, NY. (**a**) 6-h differences in rawinsonde-observed profiles of water vapor mixing ratio (blue) and potential temperature (orange) for 1200 to 1800 UTC (solid) and 1800 to 0000 UTC (broken). (**b**) is analogous to (**a**) but shows WRF-simulated profiles instead of rawinsonde-observed profiles. Mixing ratio values are multiplied by 1000.

These observed changes in thermal and moisture profiles provide evidence of the airstream sectors traversing the landfall region at one location north of the storm center. However, to better depict the spatial variation in temperature and moisture, WRF simulation results are examined along vertical cross-sections and along near-surface transects that are approximately aligned with the west-northwestward propagation direction of the surface seclusion boundary. The cross-sections chosen to be examined in detail are those that extend east-southeastward across the dropsonde site and the pair of oceanic buoys, as denoted by the magenta line in Figure 5a. Vertical cross-sections of equivalent potential temperature ($\Theta_e$) and $d\Theta_e/dz$ are shown in Figure 8a–c and 8g–i, respectively. Figure 8d–f shows transects along the vertical cross-sections of near-surface $\Theta_e$ at 70 m above ground level (AGL), which are used to distinguish the 3 airstreams, as delineated by

the vertical red lines in Figure 8. Warm sector boundaries are specified as the accelerated increase in $\Theta_e$ for the warm sector's forward (western) edge and the maximum in $\Theta_e$ for its rear (eastern) edge. As occurs in secluding cyclones, Figure 8d–f exhibits a shrinking warm sector with time as the seclusion's forward edge at the surface progresses westward faster than the warm sector's forward edge. (Note that the westward progression of the warm sector is atypical because the warm sector is atypically north of the storm center rather than south of it.) Figure 8j–l shows analogous transects but of the base height of the elevated stable layer (red layers in Figure 8g–i) as determined by the lowest level at which $d\Theta_e/dz$ reaches 20 K/km.

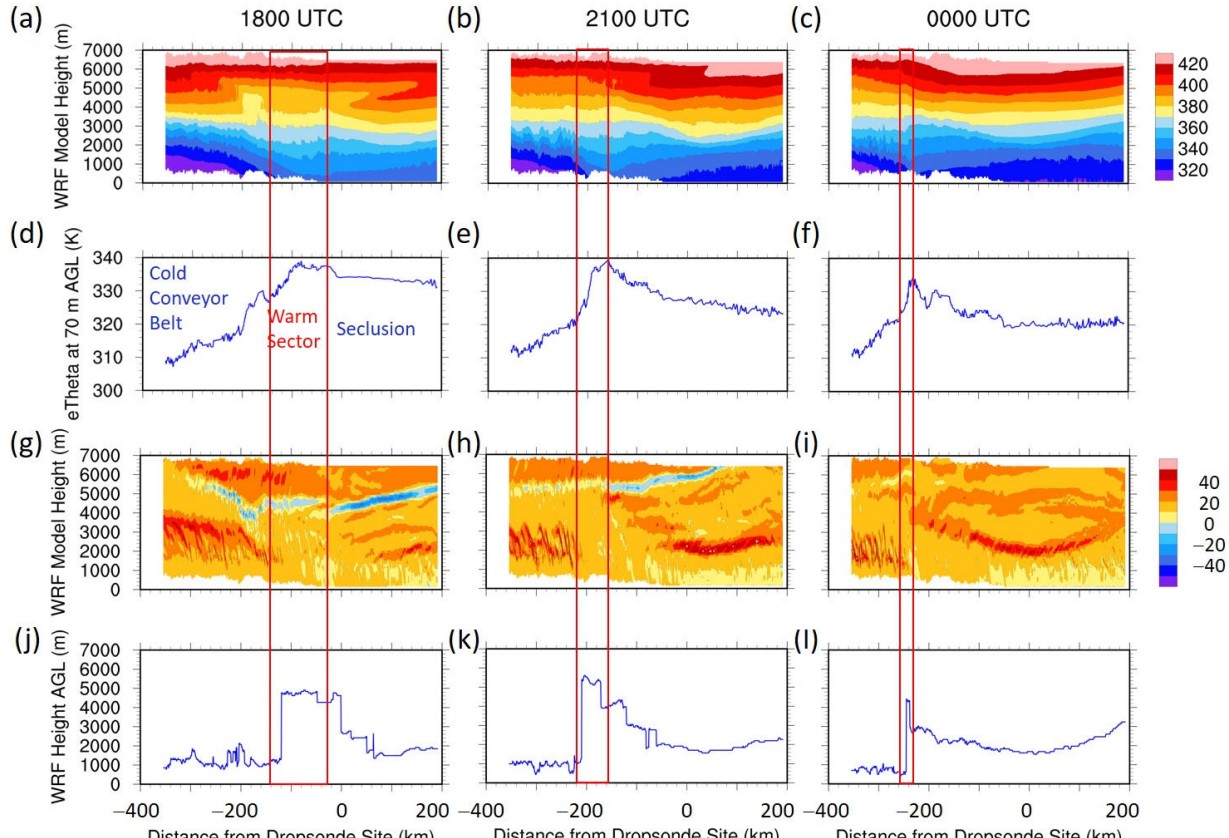

**Figure 8.** Vertical cross-sections of WRF-simulated equivalent potential temperature ($\Theta_e$) and its vertical gradient ($d\Theta_e/dz$). (**a–c**) Vertical cross-sections of $\Theta_e$ in K units along diagonal magenta line in Figure 5a, where the left end of the $x$ axis is the western end. (**g–i**) are analogous to (**a–c**), respectively, but show $d\Theta_e/dz$ in K/km units. (**d–f**) Transect plots of $\Theta_e$ in K units at an elevation of 70 m above ground level (AGL) along the cross-sections of (**a–c**), respectively. (**j–l**) are analogous to (**d–f**), respectively, but show the height of the stable layer base as determined by the height at which $d\Theta_e/dz$ reaches or exceeds 20 K/km. Vertical red lines denote the near-surface horizontal limits of the airstream sectors, which are labeled in (**d**), as determined from the 70 m AGL $\Theta_e$ transect plots of (**d–f**).

We now characterize the stable layer further because of its potentially important role in the association of rolls with specific airstreams. WRF $d\Theta/dz$ profiles calculated within the 3 separate airstreams were used to quantify stability metrics for the 3 airstreams. To this end, the $\Theta_e$ transects of Figure 8d–f were used to identify locations within each of the 3 airstreams at the 3 different times (1800, 2100, and 0000 UTC) at which WRF-simulated $d\Theta/dz$ profiles were calculated. These $d\Theta/dz$ profiles were then used to measure the maximum $d\Theta/dz$ of the capping stable layer, its elevation above ground, and the mean $d\Theta/dz$ of the lowest 1 km layer, as displayed in Table 2 as average values over the 3 times. Although the maximum $d\Theta/dz$ of the capping stable layer does not vary much among

airstream sectors, its elevation in the warm sector is 2–3 times higher than in the 2 cold airstreams. The possible importance of this difference in height among airstreams on roll presence therein is discussed below.

**Table 2.** WRF-derived stability metrics of layers below 6 km AGL for different airstream sectors.

| Airstream Sector | Cold | Warm | Seclusion |
|---|---|---|---|
| Elevation of maximum $d\Theta/dz$ in capping stable layer (km) | 1.5 | 4.9 | 2.5 |
| Maximum $d\Theta/dz$ of capping stable layer (K/km) | 11.8 | 11.7 | 10.6 |
| Mean $d\Theta/dz$ of lowest 1 km layer (K/km) | 5.3 | 4.5 | 2.4 |

As manifested by the prior observations and particularly by the WRF simulation results shown in Figure 8g–i and Table 2, locations that are traversed by the 3 different airstreams experience the following transitions. While initially within the CCB, the overrunning warm, moist air in advance of the surface warm front strengthens the overlying stable layer that caps the CCB. After the surface warm front passes and the warm, moist air extends to the ground, the lower-level stable layer becomes fully eroded. Then, after the surface seclusion boundary passes, the seclusion airstream undercuts the warm sector air and reforms a lower-level stable layer that caps the seclusion airstream.

## 4. Discussion

### 4.1. Roll Location in Other Tropical Cyclones

Since a key aspect of this work is to document and understand the location of roll vortices in Post-TC Sandy, we first describe where rolls were observed in other TCs. Note-worthy, however, is that none of the other TCs cited below were undergoing extratropical transition, as was Sandy. The earliest operational radar (WSR-57 and WSR-88D) observations of HBL rolls were reported in 1998 by Gall et al. [2] within all sectors of three intense hurricanes: Hugo, Andrew, and Erin. The rolls are described as "small-scale spiral bands" and indeed have the appearance of such, in that they are curved rather than linear, as are the features that are reported as HBL rolls since then. The majority of rolls observed in other TCs occurred in the forward sector (relative to storm motion) of the storms [1,3,12–15,17], with most of those occurring in the right-front quadrant, which is consistent with the analysis and predictions of Gao and Ginis [48].

Two of the TCs that exhibited rolls were exceptions to the location characteristics of most TCs, in that the rolls were observed in their left-rear quadrants. Both of these TCs were typhoons moving west-northwestward in the western Pacific basin. These two exceptions are Typhoon Kalmaegi which made landfall near Hong Kong in 2014 [17], and Typhoon Keith, which passed near but northeast of Guam in 1997 [3,12]. The only prior observation of rolls in an extratropical cyclone was for windstorm Thomas on 23 February 2017 in southwestern Germany [26], where the roll signatures were observed about 500 km southeast of the intense cyclone, which was centered over the North Sea. Rolls were observed in high-speed west-southwesterly wind bands of its warm conveyor belt in advance of the southwestward-trailing cold front.

Finally, it should be noted that the instruments used to observe rolls are mostly located on or near land, with the storms approaching land. Thus, this might introduce a bias for observing rolls in a storm's forward sector compared to its rearward sector, even though the forward sector is indeed where rolls are predicted to grow [48].

### 4.2. Parameters Driving Roll Growth and Size

We now lay the groundwork for hypothesizing why roll presence varies with the vertical structure and extratropical features described above. We begin by analyzing the temporal evolution of the roll growth and size parameters observed during the extended landfall period of 1500 UTC 29 October 2012 through 0600 UTC 30 October 2012. As found by Gao and Ginis [48], the primary roll growth parameter is the magnitude of positive

radial wind shear in the HBL, and the roll size parameter is the positive radial wind shear layer depth (SLD), which is also a secondary parameter for roll growth. Both parameters are associated with the surface-based inflow layer of a TC's HBL.

Time series of hourly inflow layer metrics derived from observed radar VAD wind profiles at Fort Dix are shown in Figure 9. The inflow layer top is the height at which inward velocity reaches zero, SLD is the depth over which radial wind shear is positive, and the heights of maximum inward velocity and maximum positive radial wind shear are self-explanatory. The areal extent over which the radar VAD winds were measured at Fort Dix is shown by green ovals in Figure 1. Warm sector presence at the radar site is delineated by the vertical red lines in Figure 9, which is measured from temperature time series at NJ Mesonet stations near the radar site. Warm sector arrival is specified as the time of accelerated temperature rise and its departure as the time of maximum temperature.

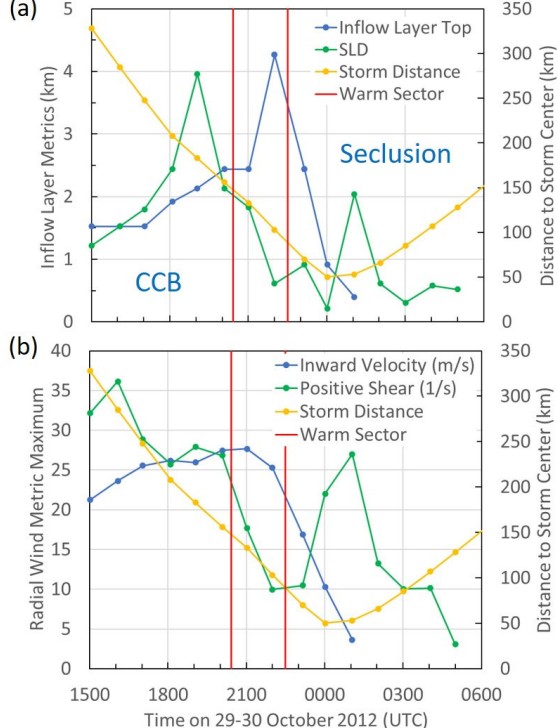

**Figure 9.** Evolution of observed inflow layer metrics obtained from Fort Dix WSR-88D radar VAD wind profiles. (**a**) Hourly time series of inflow layer top (blue), shear layer depth (SLD in green), and distance of storm center from radar observation site (yellow). (**b**) Hourly time series of the maximum inward velocity in m/s units (blue), maximum positive radial wind shear in s$^{-1}$ units (green) and distance of storm center from radar observation site (yellow). Vertical red lines in (**a**,**b**) bracket the time period during which the warm sector occupied the radar observation site at Fort Dix, as measured from temperature time series at NJ Mesonet stations near the radar site. Inflow layer metrics plotted on (**a**,**b**) are defined in Section 4.2.

Roll growth parameters observed over the Fort Dix radar site, SLD and positive radial shear, shown in green in Figure 9a,b, respectively, are both large during 1800 through 2000 UTC when the CCB occupies the radar site, but both drop rapidly during the warm sector's approach and residence over the radar site, which is consistent with the observed roll occurrence behavior. These parameters slowly increase again within the secluding airstream, but not to the magnitudes prior to the warm sector residence. The inflow layer top peaks sharply within the warm sector (Figure 9a), which is consistent with WRF results shown in Figure 8j–l and Table 2, and with roll absence in the warm sector, as elaborated upon below. The maximum inward velocity is large and continues to increase gradually

within the CCB (Figure 9b) but begins to decrease in the warm sector and continues to do so within the seclusion.

The above analysis provides evidence that the roll growth and size parameters diminish within the warm sector, as does roll presence (compare green plots of Figures 4a and 9a,b). The roll parameter analysis, however, is limited to a single site, the radar site. By using SLD calculated from the WRF simulation, one can compare it with the observed roll presence metric to provide a region-wide geographical comparison. To that end, the observed SDRV and WRF-simulated SLD are shown in Figure 10. Figure 10a–d shows the geographical distribution of SDRV at four different times at 3-h intervals, where the storm center is shown by a black dot and 50-km radar range circles are shown by grey lines. Figure 10e–h shows the geographical distribution of SLD calculated from the WRF output at almost the same four times as for Figure 10a–d.

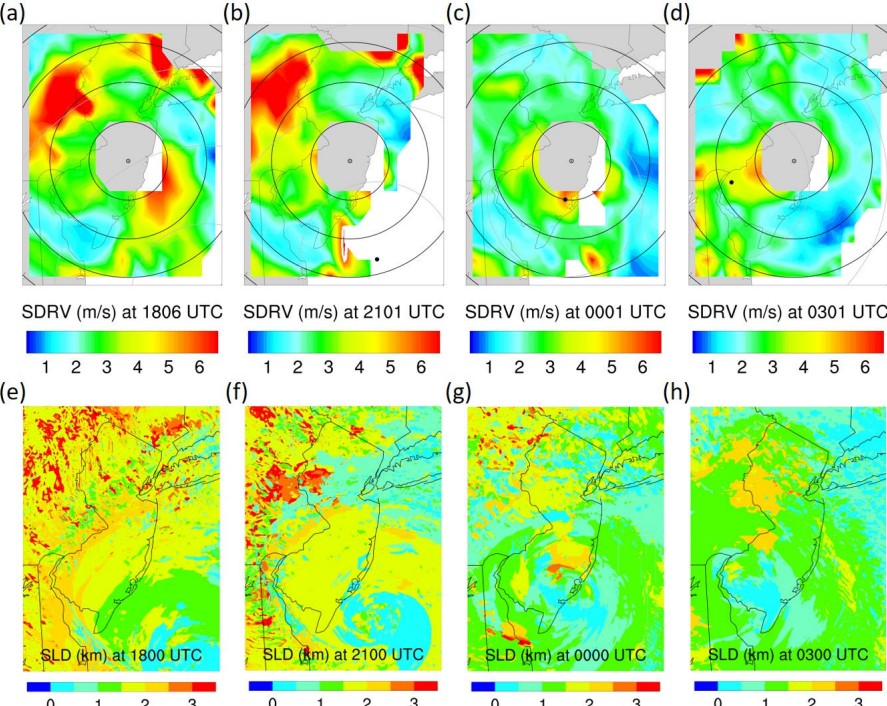

**Figure 10.** Geographical comparison of observed roll strength metric with WRF-simulated SLD. (**a**–**d**) Filled contour maps of the observed roll strength metric, SDRV, at 1806, 2101, 0001, and 0301 UTC, respectively, where the latter two times are on 30 October 2012. (**e**–**h**) Filled contour maps of the WRF-simulated roll size and growth parameter, SLD, at 1800, 2100, 0000, and 0300 UTC, respectively.

The results demonstrate similar geographical distributions and time evolution of the observed rolls to the WRF-simulated roll growth and size parameter, SLD, i.e., the largest SDRV values in Figure 10a–d are mostly observed in regions and at times when simulated SLD values in Figure 10e–h are largest. In particular, note that the blue patch of very low SDRV that progresses westward into northern New Jersey at 2101 UTC (Figure 10b) corresponds to a similar blue patch of very low SLD near the same time and location (Figure 10f), which correspond, in turn, to the time and location of the warm sector's presence. Note thereafter the increase, albeit small, over northern New Jersey of both the observed roll signal (Figure 10c,d) and simulated SLD (Figure 10g,h), which correspond to the seclusion's presence over northern New Jersey.

The geographical similarity of the observed SDRV and simulated SLD fields exhibited in Figure 10 at each time suggests that a correlation between the two variables should be demonstrable. For that purpose, SDRV and SLD are calculated for a matrix of identically-sized ($0.5 \times 0.5°$ latitude/longitude) boxes that lie within the northern half of the map domain shown in Figure 10. The limitation of the analysis to north of 40 degrees latitude is

to reduce the possible effect of terrain elevation and roughness on roll presence, since the terrain characteristics change significantly at 40 degrees latitude within the map domain, being mostly Atlantic coastal plain to the south and mostly Piedmont to the north. The standard deviation of radar radial velocity is calculated for each radar scan increment of radius and azimuth that falls within a given box, while SLD is averaged for each WRF grid point that falls within that same box. This is repeated for each 0.5 × 0.5° box that falls within the specified map domain.

Figure 11 shows that SDRV and SLD are modestly correlated early in the 6-h sequence but with a deterioration in correlation as time progresses. This is attributed to the increase in complexity of the TC's structure as it takes on more of the asymmetric features of an extratropical cyclone and the diminished matching of those features between the observed storm and the WRF-simulated storm. As is evident from Figure 8d–f, many features, such as the warm sector, become geographically smaller as extratropical transition progresses, and it is unlikely that features of this scale would match perfectly between the simulation and observed storm. Furthermore, as was noted in prior work [27], the WRF-simulated storm track begins to diverge from the actual track at about landfall time, and these differences increase with time thereafter.

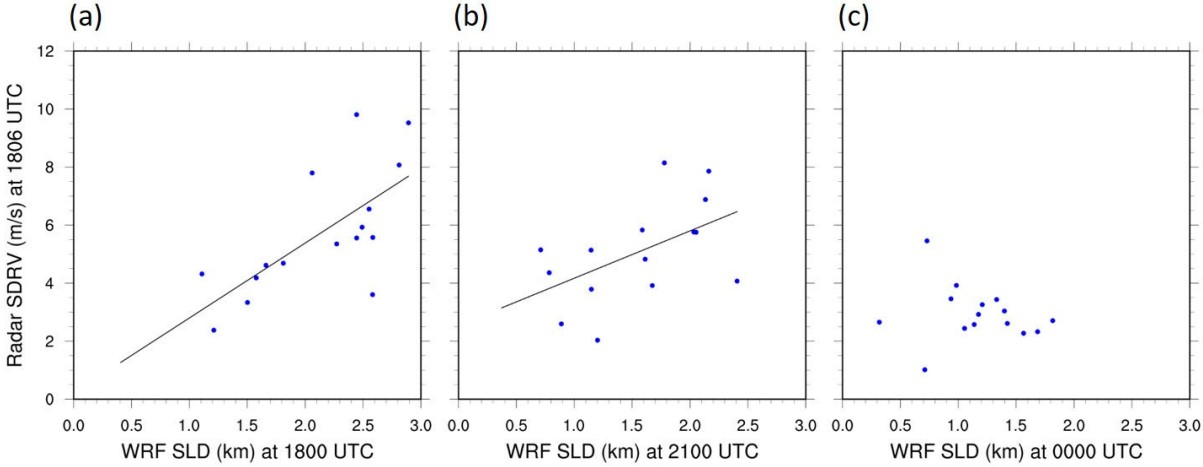

**Figure 11.** Comparison of radar-observed SDRV with WRF-simulated SLD. Both SDRV and SLD are calculated for small rectangles that are 0.5 degrees of latitude and longitude in size, wherein the average SLD is computed within the rectangle. (**a–c**) are for 1800 and 2100 UTC 29 October 2012 and 0000 UTC 30 October 2012, respectively. Black lines denote linear regression results calculated for the data points.

### 4.3. Hypothesis for Roll Location

We now hypothesize why rolls are observed only in cold airstreams. The cold conveyor belt and seclusion airstreams are capped by a stable layer at 1–2 km height AGL, with strong positive radial wind shear within those two airstreams below the stable layer. There is an abrupt rise in the stable layer base after passage of the surface warm front (Figure 8g–l) which abruptly deepens the inflow layer (Figure 9a) and, thus, significantly and abruptly reduces its radial wind shear magnitude in the warm sector (Figure 9b). As the seclusion advances through the warm sector, a weaker capping stable layer redevelops over it at 1–2 km height AGL. This allows positive radial wind shear magnitude to restrengthen within the seclusion, thus reenabling roll regrowth, albeit not as vigorously as within the cold conveyor belt (green plot in Figure 4a). This effect is manifested in WRF-simulated cross-sections and transects across airstreams, as illustrated in Figure 8g–l, by changes in the height of the elevated stable layer across those airstreams. The effect is also implicitly manifested in the observed temporal evolution of the top of the inflow layer measured at Fort Dix, as shown by the blue plot in Figure 9a.

The above discussion suggests that a relatively deep HBL can support the growth of large rolls through its ability to support a deep, intense radial wind shear layer. However, if

the HBL depth exceeds a certain depth, the magnitude of radial shear in that layer may be incapable of becoming exceedingly large. This suggests that there might exist an optimal depth of the HBL for large roll vortices to grow and survive. Therefore, future work will explore whether such an optimal inflow layer depth for promoting the growth of large rolls indeed exists and whether it can be quantified. Future work will also study the sensitivity of roll growth and size parameters, positive radial wind shear magnitude and SLD, to wind and thermal profiles within the different airstreams over northern New Jersey. It will also evaluate the potential impact of terrain elevation and roughness on roll occurrence.

**5. Conclusions**

This work builds upon earlier work [27] that reported and characterized the large roll vortices that occurred in Post-tropical Cyclone Sandy's boundary layer during landfall on New Jersey. It uses observations, supplemented by a WRF simulation, to help understand why large rolls were prevalent in specific extratropical features of a TC during its extratropical transitioning phase. An observational roll-presence metric was developed to objectively characterize the geographical distribution of roll vortices across the landfall region. Since Post-TC Sandy's extratropical characteristics were rapidly becoming more pronounced during landfall, an airstream sector analysis was done, and it was found that large roll vortices were nearly exclusively confined to cold airstreams. Rolls occurred in both the cold conveyor belt in the northwestern storm quadrant and the secluding airstream in the northeastern quadrant but not in the intervening warm sector.

Observations and WRF simulation results both indicate that the HBL is much deeper in the warm sector than in the cold airstreams. Given that Gao and Ginis [48] found that roll growth is primarily controlled by the magnitude of positive radial wind shear and that roll size is controlled by the depth of the positive radial wind shear layer, it is hypothesized that strong and deep positive radial wind shear in the cold airstreams promoted growth of large rolls therein. It is further hypothesized that the excessive depth of the warm sector diminished its ability to maintain as large a positive radial wind shear magnitude as within the cold airstreams, thus, limiting roll growth and survival within the warm sector. Thus, the presence of large rolls in Post-TC Sandy is attributed to its HBL having an optimal inflow layer depth, which is deep enough below the capping stable layer to accommodate a strong positive radial wind shear to promote roll growth, yet not so deep as to limit positive radial wind shear magnitude. Future work will more rigorously evaluate and seek to prove this hypothesis.

**Funding:** This research received no external funding.

**Data Availability Statement:** Datasets for this research are available in these in-text data citation references: NOAA Radar Operations Center (1991) [28], NOAA Earth System Research Laboratories (ESRL) Radiosonde Database (2020) [29], NOAA National Hurricane Center (NHC) Aircraft Reconnaissance Data (2020) [30], Rutgers New Jersey Weather Network (2013) [31], NOAA National Data Buoy Center (2016) [32] and Schiavone and Johnsen (2020) [47].

**Acknowledgments:** The author is grateful to Kun Gao for insightful discussions which led to the continuing pursuit of this work and to David A. Robinson and Mathieu R. Gerbush for providing data from the Rutgers New Jersey Weather Network as well as for assistance in analyzing it. The author also thanks Peter J. Johnsen for providing access to the Hurricane Sandy WRF simulation and for guidance in using it. The author greatly appreciates the thoughtful recommendations of the reviewers which helped to improve the manuscript.

**Conflicts of Interest:** The author declares no conflict of interest.

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
