# Peer review of "Airstream Association of Large Boundary Layer Rolls during Extratropical Transition of Post-Tropical Cyclone Sandy (2012)"

_2674-0494, doi:10.3390/meteorology2030022_

Round 1

Reviewer 2 Report

Overall the paper is well written. It is not clear how to match WSR-88D Doppler radar, 6-hour rawinsonde, dropsondes from aircraft reconnaissance flights, 5-minute observations from the Rutgers NJ Weather Network, and hourly oceanic observations from the NOAA National Data Buoy Center with each other. Explain in detail.

Author Response

Thank you for reviewing my manuscript. I fully understand the reviewer’s question regarding matching the different data sets because of the wide disparity in the many different types of observations used herein. My best idea on how to clarify the distinctions among the different data types and how they compare with each other was to create a table that characterizes as many aspects of the individual data types as possible, all in one place. Therefore, I inserted a table comparing the horizontal, vertical and temporal ranges and the geographical and temporal resolutions of each observational data set. I inserted Table 1 at the end of subsection 2.1. Observations. So Figure 1 and Table 1 taken together should provide an improved comparison of all observation data sets used.

Reviewer 3 Report

This paper does address a relevant topic, in that the roll vortices observed over land in ExtraTropical storm Sandy have not previously been identified in such storms over land. Sub-grid scale disturbances like these need to be properly described in the literature because as model resolutions get better, they need to account for those structures and vortices. Further, horizontal roll vortices have been theorized as being important for tornadogenesis, and hurricanes are known to occasionally spawn such tornados that are so shallow radar misses them, but they cause local enhancements to damage. References cited are appropriate, and the conclusions are accurate. Figures are fine.

Author Response

Thank you for reviewing my manuscript.

Round 2

Reviewer 1 Report

The author has addressed my comments from the last round of reviews and strengthened the manuscript. I recommend accept.